# FEDformer-Based Paddy Quality Assessment Model Affected by Toxin Change in Different Storage Environments

**DOI:** 10.3390/foods12081681

**Published:** 2023-04-18

**Authors:** Zihan Li, Qingchuan Zhang, Wei Dong, Yingjie Liu, Siwei Wei, Min Zuo

**Affiliations:** National Engineering Research Centre for Agri-Product Quality Traceability, Beijing Technology and Business University, Beijing 100048, China

**Keywords:** paddy quality change prediction, paddy, paddy storage, prediction, FEDformer

## Abstract

The storage environment can significantly impact paddy quality, which is vital to human health. Changes in storage can cause growth of fungi that affects grain quality. This study analyzed grain storage monitoring data from over 20 regions and found that five factors are essential in predicting quality changes during storage. The study combined these factors with the FEDformer (Frequency Enhanced Decomposed Transformer for Long-term Series Forecasting) model and k-medoids algorithm to construct a paddy quality change prediction model and a grading evaluation model, which showed the highest accuracy and lowest error in predicting quality changes during paddy storage. The results emphasize the need for monitoring and controlling the storage environment to preserve grain quality and ensure food safety.

## 1. Introduction

Paddy is a staple food for more than half of the world’s population and is mainly produced and consumed in Asian countries [1]; thus, its quality assurance is crucial for people’s health. However, grains such as paddy are susceptible to contamination by fungal toxins during storage [2], which affects the quality of paddy [3], among which Fusarium and Aspergillus are the main pathogenic fungi affecting the quality of paddy, and their growth is closely related to the environment [4]. For example, in suitable storage environments, such as warm and humid, the levels of harmful toxins due to fungal growth are significantly higher, such as AFB1 (Aflatoxin B1) produced by Aspergillus flavus and DON (Deoxynivalenol) produced by Fusarium oxysporum, which resulted in reduced quality and head rice yield of paddy [5,6]. The harmful toxins produced by these fungi can endanger people’s health, with aflatoxin AFB1 produced by Aspergillus flavus considered to be the most carcinogenic class of mycotoxins, causing serious health risks to humans and animals [7]; people accidentally consuming crops contaminated with vomitoxins can lead to symptoms such as poisoning and vomiting [8]. It was shown that temperature and water activity are important environmental factors affecting the growth of pathogenic fungi and mycotoxin production in paddy [9]. Lutz et al. [10] used techniques such as wireless sensor network prototypes and neural network algorithms to predict the grain quality of corn stored in silos and raffia bags under different conditions, and experiments showed that storage time had an effect on grain quality decline for all factors. Therefore, it is crucial to maintain a proper storage environment to preserve the quality of paddy grains during storage. In addition, AFB1 and DON are common and important toxins in paddy grains that pose a serious threat to grain quality and human health. Since other common non-toxic fungi rarely exceed the limit values in historical testing records, in this study, temperature and water activity were selected as environmental variables, and AFB1 and DON were chosen as influencing factors and used in a rice quality assessment model of rice grains affected by fungal changes over time in different storage environments, with regular sampling and testing.

As the world population continues to grow and global grain production increases year by year, grain quality becomes increasingly important. According to studies, grain losses during storage account for 15–20% of the total, with damage by molds causing at least 10% of grain production loss per year in some countries [11]. Therefore, improving food quality and reducing losses during the storage phase by controlling storage conditions is an urgent need that must be addressed globally to create a sustainable and resilient agri-food system.

To address this challenge, researchers conducted studies to predict and assess the quality changes of stored grains such as paddy. In previous studies, Coradi et al. [12] evaluated the seed quality of soybean cultivars under different storage conditions and experimentally showed that the best choice for storing soybean seeds for more than 6 months was composite packaging in a natural environment. Sun et al. [13] evaluated the mechanism of preservation of paddy storage in a controlled atmosphere with high CO_2_ concentration based on mass analysis and a new approach using molecular modeling tools; Shu et al. [14] designed several different experimental storage conditions in order to study the effect of storage temperature on paddy, and eventually demonstrated that sequential changes in high and low temperature conditions can affect the intermediate quality of stored paddy. The novel method based on SRC values by Li et al. provides an improved method for assessing the freshness of stored paddy [15]. Lu et al. [16] bred hybrid rice with temperature-tolerant restorative lines that not only ensured high yields through crossbreeding but also produced excellent grain quality. In order to determine the quality changes in rice during storage, Wang et al. used metabolomics to study the comprehensive metabolomic profile of typical japonica and typical indica rice varieties, and this study provided insight into the mechanisms of quality deterioration of rice during storage [17]. Jaques et al. [18] proposed a device consisting of a mechanical portable sampler with hardware device and DHT22, MHZ-14 sensors7 to predict the physical mass of 11 soybean masses during bulk transportation to avoid their mass loss.

In recent years, time series analysis was often used to reveal the development and change pattern of a phenomenon or to portray the intrinsic quantitative relationship between a phenomenon and other phenomena. The existing studies in time series forecasting can be categorized into three groups: traditional linear regression methods [19,20], machine learning methods [21,22], and deep learning methods [23,24,25,26,27,28,29,30,31,32,33,34,35,36]. Traditional models include moving average (MA), autoregressive (AR), and autoregressive moving average (ARIMA) models, all of which are widely used for time series forecasting [19,20]. Coradi et al. [1] developed six linear regression models to predict grain storage quality and evaluated the models to achieve high prediction accuracy; André et al. [37] used machine learning methods such as artificial neural networks, decision tree algorithm REPTree, and random forest to predict the quality of soybean seeds for decision making in the seed storage process. Jaques et al. [37] evaluated decision tree algorithms (e.g., REPTree and M5P), random forests, and linear regression in predicting the physical and physiological quality of soybean seeds. The results showed that these algorithms outperformed linear regression in terms of accuracy. Kumar et al. [38] used machine learning classifiers such as support vector machine (SVM) and principal component analysis (PCA) to process rice plants to identify plants as healthy or unhealthy. This method replaces the manual analysis of plant health and reduces costs. Sampaio et al. [39] used artificial neural networks (ANN) for simultaneous and quantitative analysis of paddy quality. As plant diseases cause cereal crop failure, Daniya et al. designed techniques for detecting plant diseases using deep recurrent neural networks (Deep RNN), and the experimental results showed the superior performance of RWS-based deep recurrent neural networks [40]. Ge et al. [41] used long short-term memory networks (LSTM) to predict grain storage temperature and, thus, reduce grain loss. However, these models have limitations in accurately predicting the quality changes that occur during grain storage, which are closely related to temperature and humidity. Sabanci et al. [42] predicted the yield loss due to the negative effect of wheat seeds on flour quality, caused by incorporating a long and short-term memory network (Bilstm) model in the framework, and the accuracy of the experimental results was significantly improved. Zhou et al. [43] used the Transformer model for wheat spikelet detection and, thus, yield estimation. The Transformer-based method proved to be effective for wheat spike detection under complex field conditions.

In this research, our scientific hypothesis was that storage time and environmental conditions are determinants of fungal contamination. We analyzed grain storage monitoring data from more than 20 regions to predict the quality level of paddy under different storage conditions, where the quality level was divided into vomitoxin produced by Fusarium deoxidans and aflatoxin content produced by Aspergillus flavus. We used the FEDformer method to establish a prediction model for quality changes in the paddy storage process and the K-Medoid algorithm to establish a grading evaluation model for quality changes in the paddy storage process. Specifically, controlling storage conditions can reduce mycotoxin contamination of paddy to a certain extent, and so, we judged the quality of paddy by predicting the content of mycotoxins and used three key factors, i.e., time, water activity, and temperature, to predict the quality changes during paddy storage, with the aim of reducing the uncertainty of the prediction model. Among them, the limitation ranges of water activity and temperature were 0.92–0.98 and 10–30 °C, respectively. In addition, we set an evaluation index P, which combined the current and predicted values, and combined the results of the prediction model to evaluate the quality changes of grain storage in a graded manner, where the index P combined the toxin content, as well as the current and predicted values. The experimental results showed that our prediction model exhibited the highest prediction accuracy and the smallest prediction error compared with other models, and our clustering model had the highest equivalence coefficient compared with other clustering models and was used to classify the quality changes in paddy storage into three grades. Our study can provide model and data support for paddy storage and provide meaningful reference for controlling paddy storage conditions.

## 2. Materials and Methods

### 2.1. Materials

#### 2.1.1. Data

The paddy storage monitoring data in this study were obtained from Northeast China, with cultivars such as late indica paddy. The moisture level of rice at harvest was around 14% and subsequently needed to be dried before entering the bins for storage. For sampling, we stored paddy from nine shallow round bins, and 11 points were taken from one bin for mixed testing using the cuttings method, with regular sampling in a one-day cycle. In total, 99 samples were taken, with 2970 pieces of data. The microbial toxin limits selected in this paper were as follows: aflatoxin B1 limit of 10 μg/kg and deoxyribonucleic acid deoxyribonucleic acid limit of 1250 μg/kg for paddy. In addition, the data set in the experiment was divided as shown in Table 1. In addition, the proportions of training set, test set, and validation set divided in the experiment were 70%, 20%, 10%, 2079 samples in training set, 594 samples in test set, and 297 samples in validation set, as shown in Table 1.

#### 2.1.2. Experimental Environment

In this study, the FEDformer model was built using the deep learning framework PyTorch [44], a Python-based scientific computing library that provides highly flexible deep learning tools that support dynamic computational graphs and static computational graphs. The experimental environment and parameter settings are shown in Table 2 below.

### 2.2. FEDformer-Based Paddy Quality Prediction Model

Although Transformer-based methods significantly improve the state-of-the-art in long-term series forecasting, they are not only computationally expensive but, more importantly, fail to capture a global view of the time series (the overall trend). To address these issues, FEDformer [45] combines Transformer with seasonal trend decomposition methods [46], where the seasonal trend decomposition method captures the global profile of the time series, while Transformer captures the more detailed structure. The FEDformer model combines Transformer and seasonal trend decomposition, and the main architecture uses an encoder-decoder structure with innovations. The main structure of FEDformer mainly consists of four sub-modules: Frequency Enhanced Block, Frequency Enhanced Attention), Period-Trend Decomp (MOE Decomp), and Forward Propagation (Feed Forward). The structure of the FEDformer model is shown in Figure 1.

In the encoder, the input signal was decomposed through two MOE Decomp layers, and the result of the decomposition contained both seasonal and trend components. In this process, the trend component is discarded, while the seasonal component is passed to the next layer for further learning and is finally fed to the decoder.

The input to the decoder is also subjected to three MOE Decomp layers, and is likewise decomposed into two components, seasonal and trend. The seasonal component is passed to the next layer for further learning, while the frequency-enhanced attention (FEA) layer performs frequency domain correlation of the seasonal items of the encoder and decoder. In this study, the seasonal items were learned by the frequency-enhanced attention (FEA) layer, and the trend components were summed and finally added back to the seasonal items to recover the original time series.

Through such an encoding and decoding process, the model can better separate the seasonal and trend components, and, thus, predict the trend and periodicity of the time series more accurately.

The specific function of the encoder is shown in the following equations:(1)$$\begin{array}{c} S_{en1},\_=MOEDecomp\left(FEB\left(X_{en0}\right)+X_{en0}\right) \end{array}$$
(2)$$S_{en2},\_=MOEDecomp\left(FeedForward\left(\left(FEB\left(X_{en0}\right)+X_{en0}\right)\right)\right)$$

The specific function of the decoder is as Equations:(3)$$\begin{array}{c} S_{de1},T_{de1}=MOEDecomp\left(FEB\left(X_{de0}\right)+X_{de0}\right) \end{array}$$
(4)$$S_{de2},T_{de2}=MOEDecomp(FEA\left(S_{de1},{LayerNorm(S}_{en2}\right)+S_{de1}))$$
(5)$$S_{de3},T_{de3}=MOEDecomp\left(FeedForward\left(\left(S_{de2}\right)\right)+S_{de2}\right)$$

The MOE Decomp module is composed of a set of averaging filters of different sizes that extract multiple trend components from the input signal and will generate a set of weights related to the data that will combine these trend components into a final trend line. This module decomposes the time series into a periodic term (seasonal, *S*) and a trend line (trend, *T*). It is important to note that this decomposition is not performed only once, but several times in an iterative decomposition. By using an averaging filter, the MOE Decomp module can effectively eliminate noise and outliers from the time series, resulting in a more accurate trend component. Additionally, the module can adaptively adjust the size of the filter to fit different time series data. This multi-component decomposition method allows the model to more accurately capture various trend information in the time series for better prediction and analysis.

The specific functions of the MOE Decomp module are shown in the following equation:(6)$$\begin{array}{c} X_{trend}=Softmax\left(L\left(x\right)\right)\times \left(F\left(x\right)\right) \end{array}$$
where *F*(*x*) is a set of average pooling filters and *Softmax*(*L*(*x*)) is the weight for mixing these extracted trends.

Frequency enhancement block (*FEB*) is a key technique used in time series analysis to efficiently extract frequency domain information and integrate it into a model, thus improving the accuracy and generalization of the model. The key step in FEB is to use the fully connected layer R as a learnable parameter and multiply it with the frequency features obtained by Fourier transform. This process can be interpreted as weighting the frequency features, so that the high-frequency features can be given greater weight, thus better capturing the detailed information in the time series. The structure of the frequency enhanced block is shown in Figure 2.

The specific functions of the frequency enhanced block module are shown in the following equation:(7)$$\begin{array}{c} Q^{\prime }=Select(Fourier(w\times x)) \end{array}$$
(8)$$FEB-f\left(q\right)={Fourier}^{-1}(Padding(Q^{\prime }⊙R))$$

Frequency enhanced attention (FEA) module is an important technique used in time series analysis to improve the accuracy and generalization of models by learning the intrinsic relationship between the encoder and decoder through cross-attention operations. In time series analysis, encoder and decoder are usually two independent parts of each other, which process the input and output sequences, respectively. The role of the frequency enhancement attention module (FEAM) is to exchange the information between the encoder and decoder and learn the intrinsic relationship between the signals of the two parts, thus improving the prediction accuracy and generalization ability of the model. Specifically, FEA performs cross-attention operations on the signals from the encoder and decoder. The cross-attention operation can be understood as matching and aligning the key information of the two signals to better learn the intrinsic relationship between the two parts of the signals. Through the cross-attention operation, FEAM can effectively extract correlations and similarities between the signals to achieve adaptive learning and prediction of the model.

The structure of the frequency enhanced attention module is shown in Figure 3.

The specific functions of the frequency enhanced attention module are shown in the following equation:(9)$$\begin{array}{c} Q^{\prime }=Select\left(F\left(q\right)\right) \end{array}$$
(10)$$K^{\prime }=Select\left(F\left(k\right)\right)$$
(11)$$V^{\prime }=Select\left(F\left(v\right)\right)$$
(12)$$FEA-f\left(q,k,v\right)=F^{-1}\left(Padding\left(=\sigma \left(Q^{\prime }\cdot {K^{\prime }}^{T}\right)\cdot V^{\prime }\right)\right)$$

### 2.3. K-Medoids-Based Model for Paddy Quality Assessment

In order to evaluate the grade of quality change during storage of paddy, we set an evaluation index *P*, which combines the past, current, and predicted values of toxin content, and the formula for the evaluation index *P* is shown in (13).
(13)$$\begin{array}{c} P=\left[{\bar{x}}_{i},x_{j},{\bar{x}}_{k}\right] \end{array}$$
where ${\bar{x}}_{i}$, i ∈ {1, 2, …, *n*} is the average of the toxin content of the previous *n* days, $x_{j}$ is the value of the current toxin content, ${\bar{x}}_{k}$ is the average of the predicted values of toxin content for the next *n* days, and *n* is the number of indicator variables.

In this study, a clustering algorithm was used to grade the quality changes of all samples, and for each sample, the mean value of its toxin content in the first *n* days, the mean value of its toxin content in the next *n* days, and the current value were collected, and the evaluation index P was constructed from these three indicators, and finally, the quality grading space was constructed based on the evaluation index P.

K-Medoids algorithm is a clustering algorithm based on distance metric. Its basic idea is to first randomly select k representative objects as the initial clustering centers, and then, calculate the distance between each remaining object and the representative object and assign them to the nearest clusters to produce the corresponding clustering results. k-Medoids algorithm differs from K-means algorithm in that its clustering centers are not calculated by the algorithm, but the actual data points in the data set are selected. This makes the K-Medoids algorithm more stable and more suitable for multidimensional data sets. During the iteration of the K-Medoids algorithm, the original centroids are replaced with a randomly selected non-centroid at each iteration and the clustering results are recalculated. If the clustering effect is improved, the replacement is retained; otherwise, the original centroid is restored. The iterative process of the K-Medoids algorithm is a greedy process, i.e., the decision is made at each step based on the current optimal clustering result. Unlike the K-means algorithm, the K-Medoids algorithm has good scalability and robustness when dealing with large-scale data. Additionally, it can better handle noise and outliers because its clustering centers are determined by the actual data points. The specific steps are shown in Figure 4.

### 2.4. Model Evaluation Metrics

#### 2.4.1. Evaluation Metrics for Predictive Models

When evaluating the performance and fit of prediction models, common evaluation metrics include mean absolute percentage error (MAPE), mean squared error (MSE), mean squared error (RMSE), mean absolute error (MAE), and symmetric mean absolute percentage error (SMAPE). Among them, MAPE is a measure of the percentage error between the predicted and true values, which tells us the magnitude and direction of the prediction error of the model; MSE and RMSE are measures of the squared error between the predicted and true values, which weight the square of the error in the calculation and, therefore, allow for a stricter penalty for larger errors; MAE is a measure of the absolute error between the predicted and true values, which is more straightforward compared to MSE and RMSE, weighting all errors equally; and SMAPE, which is a measure of the mean absolute percentage error, taking into account the proportional relationship between predicted and true values, and it can be used to evaluate forecasting models for different ranges of data values.

The formula for calculating the mean absolute percentage error is shown in (14).
(14)$$\begin{array}{c} MAPE=\frac{100\%}{n}\sum_{i=1}^{n}|\frac{{y^{\prime }}_{i}-y_{i}}{y_{i}}| \end{array}$$

The formula for calculating the mean square error is shown in (15).
(15)$$\begin{array}{c} MSE=\frac{1}{n}\sum_{i=1}^{n}{\left({y^{\prime }}_{i}-y_{i}\right)}^{2} \end{array}$$

The formula for calculating the root mean square error is shown in (16).
(16)$$\begin{array}{c} RMSE=\sqrt{\frac{1}{n}\sum_{i=1}^{n}{\left({y^{\prime }}_{i}-y_{i}\right)}^{2}} \end{array}$$

The formula for calculating the mean absolute error is shown in (17).
(17)$$\begin{array}{c} MAE=\frac{1}{n}\sum_{i=1}^{n}|{y^{\prime }}_{i}-y_{i}| \end{array}$$

The formula for calculating the symmetric mean absolute percentage error is shown in (18).
(18)$$\begin{array}{c} SMAPE=\frac{100\%}{n}\sum_{i=1}^{n}\frac{|{y^{\prime }}_{i}-y_{i}|}{\left(|{y^{\prime }}_{i}|-|y_{i}|\right)/2} \end{array}$$
where $y_{i}$, i ∈ {1, 2, …, *n*} is the true value, ${y^{\prime }}_{i}$, i∈ {1, 2, …, *n*} is the predicted value, and *n* is the number of indicator variables.

#### 2.4.2. Evaluation Metrics for Clustering Models

Silhouette coefficient is a metric used to evaluate the quality of clustering, which can measure the tightness and separation of clusters. The value of this index ranges from −1 to 1, and the larger the value is, the better the clustering effect is, and the closer to 1 means the tighter and more separated the samples are; the closer to −1 means the worse the clustering effect is. If the contour coefficient is 1, it means the clustering effect is very good, and the distance between different clusters is much larger than the intra-cluster distance. If the contour coefficient is −1, the clustering effect is very poor, and the distance between different clusters is much smaller than the intra-cluster distance. If the contour coefficient is 0, it means that the clustering effect is average, and the distance between different clusters is comparable to the intra-cluster distance.
(19)$$\begin{array}{c} S=\frac{1}{N}\sum_{i=1}^{N}\frac{b\left(i\right)-a(i)}{\max\;\{a\left(i\right),b(i)\}} \end{array}$$
where $a\left(i\right)$ is the average distance of other samples in the cluster to which i belongs, $b\left(i\right)$ is the minimum value of the average distance of samples from i to other clusters, and *N* is the number of samples.

## 3. Results and Discussion

### 3.1. Comparison of Predictive Paddy Quality Assessment Models

In order to effectively evaluate the performance of FEDformer in predicting quality changes during paddy storage, several deep learning models were selected for comparison experiments in this paper, and each model was tested and its prediction errors were analyzed and compared, each with the same ratio of training, testing, and validation sets. In this experiment, a five-fold cross-validation experiment was conducted to prevent the problem of overfitting. In the experiments, RNN, GRU, LSTM, and Transformer were used for time series prediction comparison experiments. In order to objectively evaluate and describe the performance of the above models, we used five evaluation metrics to calculate the error values of each model: MSE, RMSE, MAPE, MSPE, and MAE. In addition, the FEDformer model includes several super parameters that affect the prediction accuracy of the model, among which we found that the learning rate, the number of encoder layers, and the number of decoder layers have a significant impact on the prediction. Therefore, to find the parameter settings for the best performance of the model, we conducted several sets of comparative experiments by adjusting these parameters. During the experiments, for each super parameter, we adjusted one parameter at a time and then observed the change in loss until the optimal parameter was determined. Table 3 shows the settings of each parameter in the proposed model.

Out of the five prediction models, RNN and GRU had higher prediction errors, followed by the traditional LSTM. The prediction accuracy of LSTM and GRU 34re comparable, and the difference in their prediction errors was small. Meanwhile, the Transformer and FEDformer models had significantly higher prediction accuracy and lower prediction error compared to the other models. The FEDformer model exhibited the smallest prediction error compared with other models, with MAE, MSE, RMSE, MAPE, and MSPE values of 0.008, 0.0002, 0.01, 0.08, and 0.04, respectively, in the paddy test set experiments. These results are presented in Table 4.

### 3.2. Comparison of Clustering Models for Paddy Quality Variation

In this paper, to perform a cluster analysis of paddy grain quality variation, we chose the P-value of the evaluation index for each sample per day as the clustering feature. We used K-medoids and K-means algorithms to cluster the paddy grains, and the clustering results were visualized and analyzed to show more visually the pattern of contour coefficient results corresponding to different clustering algorithms and different numbers of clusters, and we plotted the histograms of the contour coefficients for each algorithm at 3 to 7 cluster numbers, as shown in Figure 3 and Figure 4. It can be seen from the plots that the contour coefficients of 3 to 7 clusters showed a decreasing trend regardless of which algorithm was used, with three clusters having the largest contour coefficients, indicating that the instances within these three clusters were more compact with each other, while the distance between clusters was larger.

In addition, the k-medoids algorithm not only had high clustering accuracy and stability, but also can effectively handle outlier and noisy data with strong adaptability and robustness, and the contour coefficient of k-medoids was the largest among the three models; so. we finally chose the k-medoids algorithm as the clustering model for paddy quality change. We divided paddy quality change into three categories, corresponding to the clustering center, quality change, and the number of samples per category for the two toxin categories, as shown in Table 2. We calculated the distance between the clustering centers and the origin based on the indicators and defined classes 1 to 3 as paddy quality change classes 1, 2, and 3, respectively. It can be seen that the indicators of cluster centers increased sequentially with the increase in quality classes.

Figure 5 shows the clustering results of the k-medoids algorithm for paddy, along with the corresponding two toxins. As can be seen from the figure, in this paper, we divided the variation of paddy quality into three categories, which had the largest contour coefficients and large distances between clusters. The clustering results of the k-medoids algorithm for paddy grains are shown in Figure 5.

Figure 6 shows the clustering results of the k-medoids algorithm for paddy, along with the corresponding two toxins. Compared with the k-medoids algorithm, we found that the contour coefficients of the clustering results of the k-means algorithm were generally lower for each cluster, but the contour coefficients of the three clusters were still the largest. The clustering results of the k-means algorithm for paddy grains are shown in Figure 6.

Table 5 shows the clustering centers, paddy mass variation, and the number of samples per level for the two toxin levels corresponding to paddy. The results showed that most of the paddy samples had deoxynivalenol DON levels at level 2, while most of the paddy samples had aflatoxin B1 levels distributed at level 1 versus level 2. The clustering centers for the two toxin levels corresponding to the paddy grains, the variation in paddy grain quality, and the samples for each level are shown in Table 5.

### 3.3. Analysis of Paddy Quality Change Results

We visualized the change in paddy quality grade based on the clustering results of k-medoids, as shown in Figure 7. According to the results, the grade of paddy grain increased slowly at lower temperature and moisture activity and rapidly at higher temperature and moisture activity, which means that the growth rate of aflatoxin and vomitoxin content increased slowly at low temperature and low moisture activity, while the growth rate of both toxins tended to increase when the temperature increased and the moisture activity increased. In addition, the grade of paddy increased to the highest grade with the increase in storage time under any storage environment conditions. The above results demonstrate that storage time and environmental conditions are determinants of fungal contamination of the paddy grains.

Based on the above observations, it is evident that changes in the environment during storage have a significant effect on the quality grade of the paddy grains. The levels of aflatoxin B1 and deoxyribonucleic acid are influenced by water activity and temperature, and their growth states both increase with water activity, with the optimum growth state occurring at a water activity of 0.98 and a temperature of 30 °C. Therefore, we recommend necessary storage measures to reduce quality changes and toxin production during paddy storage, which will help ensure the quality of paddy and the health of consumers.

## 4. Conclusions

Paddy is one of the most important food crops in the world, but quality changes during storage of paddy have been an important issue in the field of food security. Among them, aflatoxin B1 and deoxyribonucleic acid are common toxins in paddy storage. Therefore, unfavorable storage conditions can promote mold growth and toxin production, leading to the degradation of paddy quality and health risks to consumers.

In this paper, five indicators were collected using the levels of AFB1 and DON as input variables. Then, their future toxin levels during paddy storage were predicted based on the FEDformer prediction model. We constructed an evaluation index P using the current and predicted toxin levels and developed a k-medoids-based quality change model for a comprehensive classification of the quality of paddy in storage. Finally, we combined the results of the clustering with our data to visualize the variation of paddy quality grades over time under different storage environmental conditions, and the results showed that the limiting temperature and water activity during storage are key factors for the development of aflatoxin B1 and deoxyribonucleic acid as well as grain quality. Their growth states both increased with increasing water activity, and the best fungal growth state occurred at a water activity of 0.98 and a temperature of 30 °C. Necessary storage measures should be taken to reduce quality changes and toxin production during storage of paddy grains. This will help ensure the quality of the paddy and the health of the consumer.

## Figures and Tables

**Figure 1 foods-12-01681-f001:**
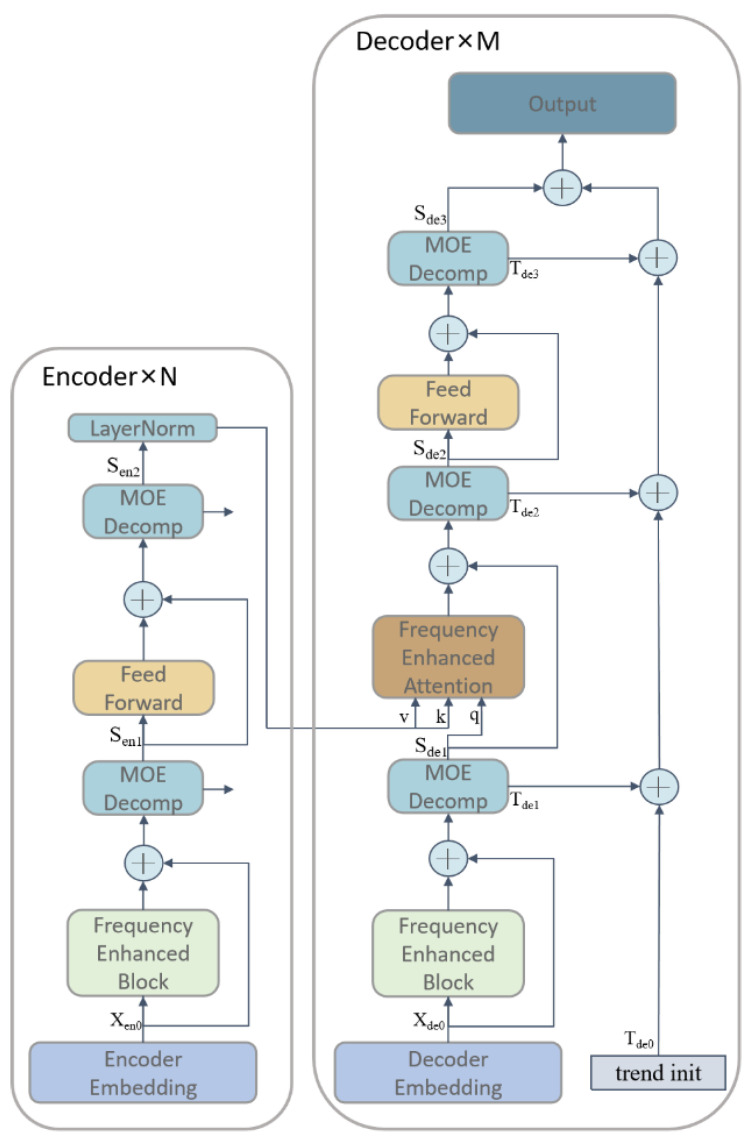
FEDformer Model.

**Figure 2 foods-12-01681-f002:**
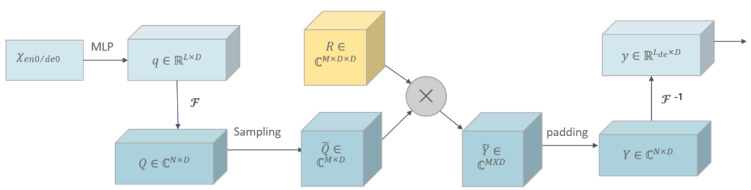
Frequency Enhanced Block.

**Figure 3 foods-12-01681-f003:**
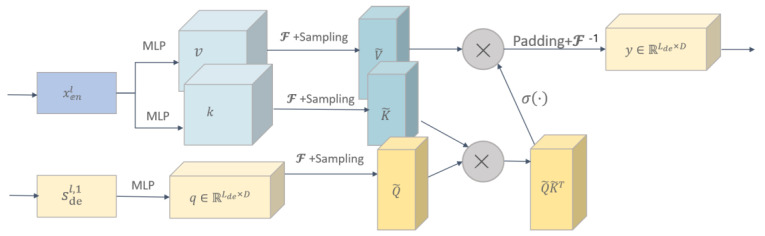
Frequency Enhanced Attention module.

**Figure 4 foods-12-01681-f004:**
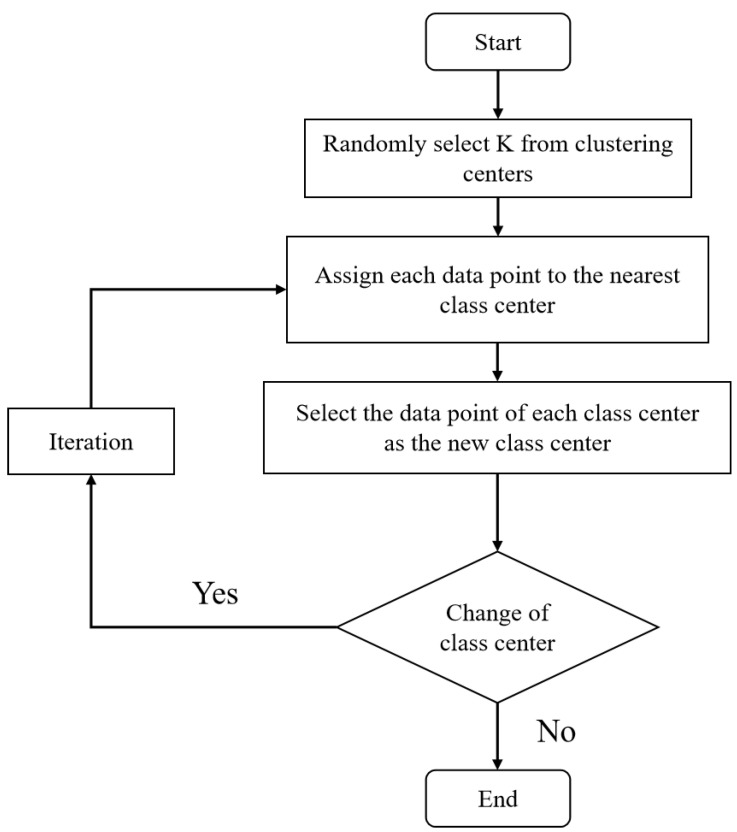
K-medoids algorithm flow chart.

**Figure 5 foods-12-01681-f005:**
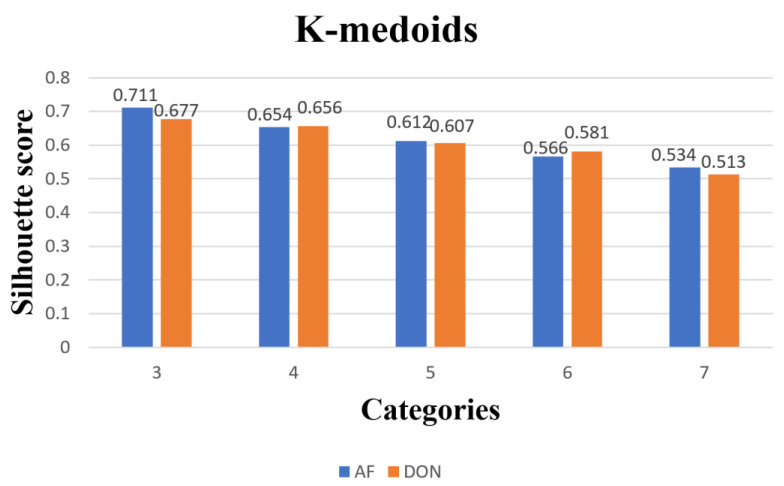
K-medoids 5 types of clustering category profile coefficients.

**Figure 6 foods-12-01681-f006:**
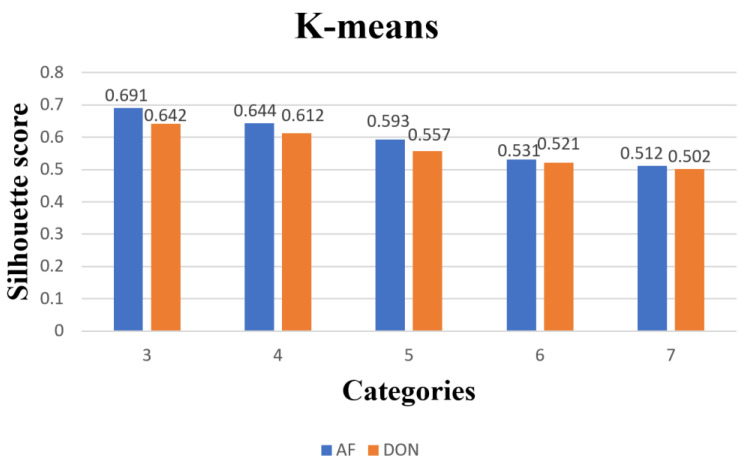
K-means 5 types of clustering category profile coefficients.

**Figure 7 foods-12-01681-f007:**
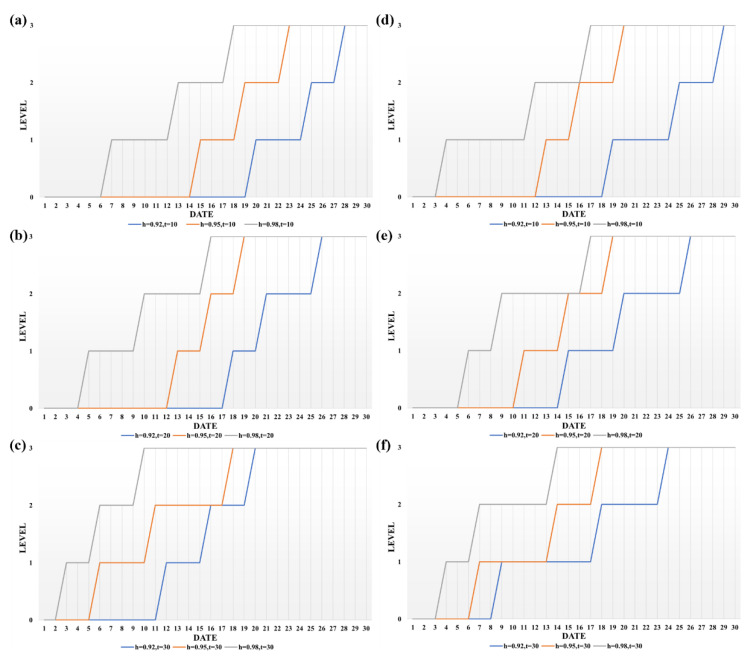
Folding line graph of the quality grade of aflatoxin and vomitoxin in paddy grains with different storage conditions. (**a**–**c**) are the line graphs of the quality grade of aflatoxin with time at 0.92, 0.95, and 0.98 water activity at 10 °C, 20 °C, and 30 °C, respectively. (**d**–**f**) are line graphs of vomitoxin quality grade over time at 0.92, 0.95, and 0.98 water activity at 10 °C, 20 °C, and 30 °C, respectively.

**Table 1 foods-12-01681-t001:** Partitioning of the data set in the experiment.

Dataset	Training Set	Test Set	Validation Set
2970	2079	594	297

**Table 2 foods-12-01681-t002:** Experimental platform and environmental parameters.

Interpreter	Programming Languages	Dependency Packages
Toolkit	Python (3.7, PSF, FDK, MD, United States)	Numpy 1.21.5
		Scikit_Learn 1.0.2
		Pandas 0.25.1
		Torch 1.12.0
		Matplotlib 3.5.2

**Table 3 foods-12-01681-t003:** Parameter setting for the FEDformer model.

Model	Learn Rate	Encoder Layers	Decoder Layers	MAPE
FEDformer	0.001	3	3	0.12
	0.001	3	2	0.10
	0.0001	2	2	0.07
	0.0001	2	1	0.04

**Table 4 foods-12-01681-t004:** Experimental results on the comparative performance of FEDformer-based paddy quality prediction models.

Model	MAE	MSE	RMSE	MAPE	MSPE
RNN	0.25	0.04	0.21	0.85	7.32
GRU	0.24	0.04	0.20	0.85	7.30
LSTM	0.21	0.03	0.17	0.83	6.40
Transformer	0.027	0.0012	0.03	0.55	2.30
FEDformer	0.008	0.0002	0.01	0.08	0.04

**Table 5 foods-12-01681-t005:** Clustering centers and ranking of the three clusters of paddies.

Categories	${\bar{x}}_{i}$	$x_{j}$	${\bar{x}}_{k}$	Sample Size	Quality Level
AFB1 1	2.6036	4.1476	5.9409	695	Level 1
AFB1 2	4.4638	6.2585	8.7218	693	Level 2
AFB1 3	6.0306	9.4696	12.6445	196	Level 3
DON 1	262.8922	321.1738	405.2554	495	Level 1
DON 2	482.1590	584.5295	685.5420	905	Level 2
DON 3	787.0462	992.1297	1143.4456	184	Level 3

## Data Availability

The data used to support the findings of this study are available from the corresponding author upon request.

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
