# Peer review of "FEDformer-Based Paddy Quality Assessment Model Affected by Toxin Change in Different Storage Environments"

_foods, 2023, doi:10.3390/foods12081681_

Round 1

Reviewer 1 Report

The manuscript “FEDformer-based paddy quality assessment model affected by fungal changes in different storage environments” brings relevant scientific contributions to the area, with the proposal of a model for assessing the quality of stored rice. The manuscript has high potential, however, some important points need to be justified and revised in the manuscript, mainly regarding storage conditions and fungal/mycotoxin contamination.

1)      In the introduction, more specific information should be presented, involving storage environments and fungal contamination in rice. Fungi species x environments? What are the limits for temperature, relative humidity, water activity? Storage time can be considered a determining factor for increased fungal contamination, regardless of the environment. What are the relationships between storage time and fungal contamination?

Jaques, L. B. A., Coradi, P. C., Müller, A., Rodrigues, H. E., Teodoro, L. P. R., Teodoro, P. E., ... & Steinhaus, J. I. (2022). Portable-Mechanical-Sampler System for Real-Time Monitoring and Predicting Soybean Quality in the Bulk Transport. IEEE Transactions on Instrumentation and Measurement71, 1-12.

2) In post-harvest (storage) the conservation of the product is sought, that is, the prevention of deterioration and loss of quality. In this sense, control must be established. When fungi and/or mycotoxins are present, the grains are already compromised and there is no way to reverse it in terms of quality. Thus, the most important aspect of this study is the control conditions to avoid contamination. The manuscript should be better addressed on the appropriate storage conditions and given less importance to contamination.

3) What were the scientific hypotheses of the experiment? Make it clear in the introduction in a sentence or paragraph.

4) By more clearly delineating these conditions described above, the proposed model may have greater meaning and importance. Please look for recent articles in the literature that bring these approaches (storage environment conditions x grain quality) to better justify the study design.

5) I suggest a review topic in the manuscript (review item) after the introduction, highlighting the points above.

6) In the material and methods, limits of toxins were used, however the focus of the study is on fungal contamination. The presence of fungi in stored grains (even under high contamination) does not mean that the grains will be contaminated by mycotoxins. For mycotoxin contamination to occur, fungi must be toxigenic, which is often not the case. By the way, the detection of mycotoxins in stored grains is very difficult due to the proportions (ppm, ppb), so many times there is contamination and they are not detected. Thus, it would be much more advantageous to control storage conditions focused on contamination rather than toxin limits. How could the authors explain this decision? Likewise, the established approach (title-fungi, introduction-fungi and toxins, material and methods-toxins) is confusing.

7) Results and discussion, further discussion of the results is expected, comparing with data from the literature, mainly under a practical view of the storage area. In the current situation, no discussion of results.

8) On the other hand, the conclusions are very broad, more like a continuation of the previous topic. The conclusions must be objective and punctual, responding objectively to the objectives proposed in the study.

Author Response

I really appreciate all your comments and suggestions! We have carefully studied your valuable comments and made every effort to revise the manuscript. In the document is my itemized response to the changes/corrections found in the resubmitted manuscript.

Reviewer 2 Report

Reviewer’s comments

The authors have used FEDformer models for based paddy quality . this is an interesting approach to model the quality parameters.

Some minor changes are needed.

Paddy to be used instead of paddy rice. Paddy is also known as rough rice.Rice is a term generally used for milled rice.

Please mention milling yield or head rice yield instead of yield.

In the introduction only the aims and objective are mentioned not the results. Last 3 lines should be deleted: line 100-104

Materials and methods:

Mention the paddy harvesting season and the variety. How much paddy was stored. How was the sampling done? L 106

The tabular representation of data and platform may be included in text. Donot include unnecessary data. You may give the software details: L 120

Table 5: use upto 4 decimal points maximum.

Tha authors have shown eth model acuuracy in tersm fo RMSE, MAE etc, but the predicted a nd experimental values for aflatoxin is not shown anywhere. How many data points or % of total data was used for, testing and validation? If the climatic conditions change what would be the aflatoxin content in paddy. One example of a severe condition may be presented. 

Author Response

(The authors gave the same response as above.)

Round 2

Reviewer 1 Report

The authors have effected all the changes suggested. The article can be accepted for publication.

Author Response

Thank you very much for taking the time to review this manuscript. I really appreciate all your comments and suggestions! We have carefully studied the valuable comments from you, the assistant editors and reviewers, and have made every effort to revise the manuscript. In the document is my itemized response to the changes/corrections found in the resubmitted manuscript.
We would also like to thank you for allowing us to resubmit a revised version of the manuscript.
